# Semi-Supervised Segmentation via Embedding Matching

**Weiyi Xie**                                                          WEIYI.XIE@STRYKER.COM
**Nathalie Willems**                                          NATHALIE.WILLEMS@STRYKER.COM
**Nikolas Lessmann**                                      NIKOLAS.LESSMANN@STRYKER.COM
**Tom Gibbons**                                                   TOM.GIBBONS@STRYKER.COM
**Daniele De Massari**                                  DANIELE.DEMASSARI@STRYKER.COM
*Stryker, 325 Corporate Dr, Mahwah, NJ 07430*

**Editors:** Accepted for publication at MIDL 2024

## Abstract

Deep convolutional neural networks are widely used in medical image segmentation but require many labeled images for training. Annotating three-dimensional medical images is a time-consuming and costly process. To overcome this limitation, we propose a novel semi-supervised segmentation method that leverages mostly unlabeled images and a small set of labeled images in training.

Our approach involves assessing prediction uncertainty to identify reliable predictions on unlabeled voxels from the teacher model. These voxels serve as pseudo-labels for training the student model. In voxels where the teacher model produces unreliable predictions, pseudo-labeling is carried out based on voxel-wise embedding correspondence using reference voxels from labeled images.

We applied this method to automate hip bone segmentation in CT images, achieving notable results with just 4 CT scans. The proposed approach yielded a Hausdorff distance with 95th percentile (HD95) of 3.30 and IoU of 0.929, surpassing existing methods achieving HD95 (4.07) and IoU (0.927) at their best.

**Keywords:** Semi-supervised Segmentation, Pseudo-labeling

## 1. Introduction

Semantic segmentation is crucial in quantitative medical imaging analysis, such as patient-specific surgical planning for total hip arthroplasty based on accurate segmentation of target structures. With the advent of deep neural networks (Ronneberger et al., 2015; Çiçek et al., 2016; Milletari et al., 2016; Isensee et al., 2021), supervised approaches to semantic segmentation have seen significant advancements. However, obtaining large-scale annotated data can be prohibitively expensive in practice. To address this challenge, many attempts have been made towards semi-supervised semantic segmentation (Yu et al., 2019; Wang et al., 2022; Tarvainen and Valpola, 2017; Vu et al., 2019; Zhang et al., 2017; Verma et al., 2022; Chen et al., 2021; Ouali et al., 2020), which learns from only a few labeled samples and numerous unlabeled ones. Consistency training is a widely-used approach in semi-supervised learning to enforce the model's outputs to be consistent for similar input data points (Tarvainen and Valpola, 2017). Consistency training methods can be classified into four types, based on where the perturbations are applied. At the input level, uncertainty-aware mean teacher (Yu et al., 2019), strong-to-weak consistency (Yang et al., 2022a), and unsupervised data augmentation (Xie et al., 2020) have applied random noises and data augmentation

techniques on input images. At the feature level, cross consistency training (CCT) (Ouali et al., 2020) enforces aligned outputs from perturbed and non-perturbed features. At the network level, Luo et al. (Luo et al., 2022) propose aligning the outputs from the CNN and transformer networks since different networks may extract varying information. At the task-level, dual-task consistency (DTC (Luo et al., 2021)) proposed to align predicted segmentation maps with the predicted distance maps from the same network with two output heads.

The other popular concept in semi-supervised segmentation is pseudo-labeling. The underlying idea is to assume that some model predictions are reliable enough to pseudo-label unknown voxels. Since model predictions may not always be reliable, several works use uncertainty (Yu et al., 2019; Wang et al., 2022) or confidence scores (Sohn et al., 2020; Yang et al., 2022b) to discard unreliable model predictions in pseudo-labeling. However, this exclusion leads to an information loss because some voxels in unlabeled images that have not been pseudo labeled are discarded as well, and therefore cannot be used in training supervision. Furthermore these voxels, probably with highly uncertain predictions from a trained network, may be areas critical for object segmentation (e.g., object boundaries). In this paper, we introduce a new strategy to pseudo-label unknown voxels with unreliable model predictions. Our method propagates labels from manual segmentations using voxel-wise embedding correspondence. This correspondence is established when the embeddings generated by the student model for the unlabeled voxels match those produced by the teacher model for the labeled voxels. We evaluated this method on a challenging collection of CT images for segmenting hip bones. We demonstrated substantial improvements of our method for semi-supervised hip bone segmentation in comparison with state-of-the-art methods (Chen et al., 2019; Ouali et al., 2020; Yu et al., 2019; Tarvainen and Valpola, 2017). Our contribution manifests in two key aspects: first, the introduction of a novel label-propagation schema through embedding correspondence; and second, an end-to-end semi-supervised segmentation framework facilitating the pseudo-labeling of all regions within unlabeled images.

## 2. Method

### 2.1. Overview

Our framework, depicted in Fig. 1, builds upon the uncertainty-aware mean teacher method (Yu et al., 2019). We use two 3D U-Net (Çiçek et al., 2016) models, the student and the teacher network, with identical architectures. To accommodate our computational budget, we reduced the number of convolution filters by half compared with the original 3D U-Net. The student network is trained using stochastic gradient descent, while the teacher model is updated by maintaining a moving average of the student network's weights.

During training, our data are partitioned into labeled and unlabeled sets, denoted by $D_L = \{(x_i, y_i)\}_{i=1}^{N_L}$ and $D_U = \{(x_i)\}_{i=1}^{N_U}$, respectively. $N_L$ and $N_U$ represent the number of labeled and unlabeled training images, respectively. The 3D input image is denoted as $x_i \in R^{H \times W \times D}$, and the corresponding binary ground truth segmentation as $y_i \in \{0, 1\}^{H \times W \times D}$, where foreground objects are hip bones (pelvis and femoral head) manually delineated on CT images (see section 3.1). For simplicity, we leave out $i$ in the following notations.

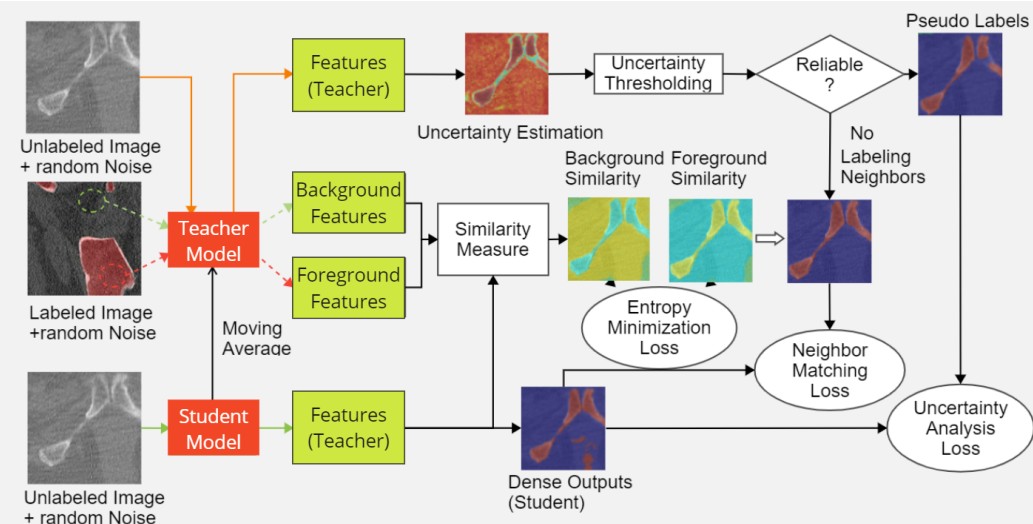

Figure 1: Semi-supervised Semantic Segmentation Framework for training with unlabeled images. The teacher model produces pseudo labels for training the student model via uncertainty analysis and nearest-neighbor matching.

We randomly sample same amount of labeled and unlabeled images for each training batch. For training the student model, the loss computed on labeled images is a combination of cross-entropy and Dice loss. For unlabeled images, the teacher network is used to provide pseudo-labels, as illustrated in Fig. 1, for computing the semi-supervised loss. To adjust the weight for each loss term, we use a Gaussian ramp-up function similar to (Laine and Aila, 2016) that gradually increases the importance of the semi-supervised loss:

$$ramp\_up(T, s) = s \cdot \exp[-5.0(1.0 - T/T_N)^2] \tag{1}$$

Here,' $s$ is a loss-specific scaling factor indicating the value at the ramp-up maturity, $T$ is the current iteration, and the $T_N$ is the total number of training iterations.

### 2.2. Pseudo Labeling via Uncertainty Analysis

Uncertainty analysis aims to identify reliable predictions ($\hat{y}_t$) from the teacher network to train the student network on unlabeled images. Without manual correction, predictions from the teacher model may be inaccurate. To select only reliable predictions, we employ an uncertainty analysis technique based on Monte Carlo Dropout (Kendall and Gal, 2017), as outlined in (Yu et al., 2019). In detail, we perform $M$ stochastic forward passes on the teacher model under random dropout and Gaussian input noise to measure predictive entropy for each image $x$. By averaging the predictive entropy at each forward pass $m$, we compute the dense uncertainty map $H_t^x$ as the following:

$$H_t^x = -\sum_{m=1}^{M} \sum_{c \in 0,1} p_t^m(y = c \mid x) \log[p_t^m(y = c \mid x)]/M \tag{2}$$

where the subscript $t$ indicates that $H_t^x$ is measured based on the teacher model's predictions and $p_t(y = c|x) \in R^{H \times W \times D}$ is the dense softmax probability map from the teacher network for class $c$ ($c \in \{0, 1\}$) given the input image $x$. The pseudo labels are produced from the teacher model predictions (i.e softmax probability map) as $\hat{y}_t = argmax_c(p_t(y = c \mid x))$. We apply the ramp-up procedure (Eq. 1) to compute the threshold for the current iteration $T$ as $\lambda_T = [0.75 + ramp\_up(T, 0.25)] \cdot ln(2)$ (Kendall and Gal, 2017) to determine which predictions of the teacher can be considered reliable. The final uncertainty analysis loss $L^{UA}$ for input $x$ at training iteration $T$ is the binary cross entropy (BCE) loss:

$$L_{UA} = \frac{\sum [1(H_t^x < \lambda_T) \cdot BCE(\hat{y}_t, CNN_s(x))]}{\sum 1(H_t^x < \lambda_T)}$$

taking the pseudo label $\hat{y}_t$ and the student model's output $CNN_s(x)$ as the inputs, where the $1(\cdot)$ is an indicator function used to mask out the unreliable predictions in the loss computation. Stronger random noise is added to the input to the student network compared to the teacher network, following weak-strong consistency training (Yang et al., 2022a).

### 2.3. Nearest Neighbor Pseudo Labeling

Uncertainty analysis involves identifying and excluding unreliable pseudo-labels generated by the teacher network (refer to Section 2.2). However, this can result in losing coverage on many bone surface areas in the unlabeled images, where the model's uncertainty is typically high. Therefore, we propose a novel label propagation technique based on voxelwise embedding correspondence to pseudo-label these regions for training.

#### 2.3.1. Embedding Sampling and Label Propagation

Given a pair $(x_L, y)$ of a labeled image and its corresponding ground truth segmentation, we first find the object surface using the ground truth segmentation $y$. Subsequently, we randomly select $k$ voxels to form two sets based on their distances from the object surface: the voxels near the surface inside the object (object voxels) and those near the surface outside object (background voxels).

We use the dense feature map (before squeezing the channels with $1 \times 1 \times 1$ convolutions at the output layer) from the teacher network to embed these voxels. We denote the embeddings for the object voxels and background voxels as $f_{L+}^t \in R^{C \times k}$ and $f_{L-}^t \in R^{C \times k}$, respectively, where $C$ is the dimension of the embedding space.

When presented with an unlabeled image $x_U$ and its corresponding voxel embeddings $f_U^s \in R^{C \times H \times W \times D}$ from the student network, we proceed to assign pseudo labels by comparing $f_U^s$ with the labeled embeddings $f_{L+}^t$ and $f_{L-}^t$ in terms of their pairwise similarities. A voxel is assigned to the object class if its corresponding embedding is more similar to the object embeddings than to the background embeddings. This is equivalent to a $k$-nearest neighbor classifier in the embedding space. Interestingly, even when the labeled embeddings are on the wrong side of the true decision hyperplane, and the teacher network mislabels their corresponding voxels, we can still utilize the label of these voxels to guide the student training of other voxels in the unlabeled images given their embeddings similarity.

We denote $K$ as the similarity measurement kernel (cosine similarity). To obtain dense similarity maps $K(f_{L+}^t, f_U^s) \in R^{H \times W \times D}$ and $K(f_{L-}^t, f_U^s) \in R^{H \times W \times D}$, we first compute

pairwise similarity scores between $f_{L\pm}^t$ and $f_U^s$ before reducing the scores across the $k$ labeled embeddings by averaging. However, to avoid the contribution from potential outliers in the embedding space, we employ an ensemble of nearest neighbor classifiers by computing the dense similarity maps $l$ times, where the results of each run are denoted as $K_l(f_{L+}^t, f_U^s)$ and $K_l(f_{L-}^t, f_U^s)$. At each run, we sample different sets of $k$ voxels to generate $f_{L+}^t$ and $f_{L-}^t$. The pseudo labels $\hat{y_{nn}}$ from the ensemble of nearest neighbor classifiers are obtained by averaging all $l$ dense similarity maps. We added supplementary materials to showcase the computation of dense similarity maps is robust against the choice of similarity measurement kernels and reduce operations.

These pseudo labels can be used to supervise training for the voxels at which the uncertainty measurements are higher than the predefined threshold, i.e., where the teacher model's predictions are not reliable enough to treat them as pseudo labels (see Sec. 2.2). We refer to this loss term as the nearest neighbor matching loss $L_{NN}$, formulated as:

$$L_{NN} = \frac{\sum[1(H_t^x \geq \lambda_T) \cdot BCE(\hat{y_{nn}}, CNN_s(x))]}{\sum 1(H_t^x \geq \lambda_T)} ,$$

$$\hat{y_{nn}} = 1[K(f_{L+}^t, f_U^s) > K(f_{L-}^t, f_U^s)] \qquad (3)$$

where 1 is the indicator function. Note that the computation of $\hat{y_{nn}}$ is cut off from the gradient computation graph to avoid the situation where the student model is involved in both back-propagation and pseudo-label generation.

### 2.3.2. Entropy Minimization For Nearest Neighbor Classifiers

Segmentation around the object surface is generally challenging, partly due to potential inconsistencies in manual labeling and blurry voxel intensities around the surface. This can lead to ambiguous model predictions and corresponding cluttered features in the embedding space. Therefore, it can be challenging to have a clear separation in dense similarity maps (as discussed in Sec. 2.3.1) such that $K_l(f_{L+}^t, f_U^s)$ and $K_l(f_{L-}^t, f_U^s)$ are distinguishable. To maximize the similarity map separation, we propose to add an entropy minimization loss term:

$$L_{EN} = -(p_\oplus \log p_\oplus + p_\ominus \log p_\ominus)$$

$$p_\oplus = e^{K(f_{L+}^t, f_U^s)}/(e^{K(f_{L+}^t, f_U^s)} + e^{K(f_{L-}^t, f_U^s)}), \qquad (4)$$

$$p_\ominus = 1.0 - p_\oplus$$

$p_\oplus$ and $p_\ominus$ are predictive probabilities from the nearest neighbor classifier assigning labels to the object and background class, respectively. These can be computed by simply applying softmax on the dense similarity maps ($K(f_{L+}^t, f_U^s)$ and $K(f_{L-}^t, f_U^s)$) concatenated channel-wise. The loss $L_{EN}$ is intended to minimize the predictive entropy of the nearest neighbor classifier and thus max-margin its classification decision boundary.

The total loss $L$ for training the student model on each mini-batch is the sum of the loss on labeled images $L_S$ and the loss on unlabeled images $L_U$. $L_U$ is the weighted summation of $L_{UA}$, $L_{NN}$, and $L_{EN}$. $L$ and $L_U$ can be formulated as:

$$L = L_S + L_U$$

$$L_U = w_{UA} \cdot L_{UA} + w_{PS} \cdot (L_{NN} + L_{EN}) \qquad (5)$$

$w_{UA}$ follows a ramp-up procedure $ramp\_up(T, 0.25)$, and $w_{PS}$ follows a ramp-up procedure $ramp\_up(T, 0.125)$. The final weights for $w_{UA}$ and $w_{PS}$ are 0.25 and 0.125, respectively.

Table 1: Results of the proposed method, compared with SOTA semi-supervised and base-line methods. Evaluation metrics include IoU and HD95 in mm. Boldface denotes the result significantly better than others (p <0.05).

| Methods | Metric | 50 | 150 | 300 | 1000 | 2500 | 6266 |
|---|---|---|---|---|---|---|---|
| Supervised | IoU | 0.894 | 0.921 | 0.921 | 0.946 | 0.947 | 0.948 |
| baseline | HD95 | 9.45 | 5.30 | 3.00 | 2.04 | 1.97 | 1.92 |
| MT(Tarvainen and Valpola, 2017) | IoU | 0.918 | 0.925 | 0.939 | 0.946 | 0.947 | - |
|  | HD95 | 6.39 | 4.91 | 2.45 | 1.99 | 1.90 | - |
| UAMT(Yu et al., 2019) | IoU | 0.926 | 0.929 | 0.939 | 0.943 | 0.944 | - |
|  | HD95 | 5.00 | 3.92 | 2.49 | 2.22 | 2.12 | - |
| CCT(Ouali et al., 2020) | IoU | 0.922 | 0.927 | 0.935 | 0.946 | 0.947 | - |
|  | HD95 | 5.72 | 4.40 | 2.87 | 1.99 | 1.90 | - |
| MTC(Chen et al., 2019) | IoU | 0.923 | 0.923 | 0.936 | 0.941 | 0.947 | - |
|  | HD95 | 4.94 | 4.55 | 2.55 | 2.18 | **1.90** | - |
| DTC(Luo et al., 2021) | IoU | 0.927 | **0.937** | 0.938 | 0.941 | 0.941 | - |
|  | HD95 | 4.07 | 3.29 | 2.49 | 2.28 | 2.18 | - |
| Proposed | IoU | **0.929** | 0.931 | **0.939** | **0.947** | **0.947** | - |
|  | HD95 | **3.30** | **2.99** | **2.45** | **1.95** | 1.94 | - |

## 3. Experiments

### 3.1. Data

To evaluate the proposed semi-supervised learning framework, we conducted experiments on a cohort of 950 subjects. Hip bones in their CT scans were manually-segmented with proprietary software. Scans were randomly partitioned into 350 for training, 100 for validation, and 500 for testing. We further preprocessed CT scans in each split into 3D patches by sampling at different locations in the image. We ensure images to have isotropic spacing of 1 millimeter and normalized intensity values using the mean and standard deviation computed using 3D patches statistics. We collected 6266 3D patches for training, 1174 for validation, and 1200 for testing, where test set patches were sampled with minimal overlaps and chances from the same subject.

For semi-supervised training, we divided the training set into five distinct subsets, each consisting of 50, 150, 300, 1000, and 2500 labeled patches, with the remaining patches being considered as unlabeled. These 3D patches were sampled from 4, 10, 21, 71, and 180 CT scans. Total 6266 training patches were sampled from 350 training CT scans. Each semi-supervised subset was named after the number of labeled patches. For instance, 'Subset 50' comprised 50 labeled patches, while the remaining 6216 patches were considered unlabeled and utilized for semi-supervised training without their reference segmentations.

### 3.2. Comparison with supervised baseline and semi-supervised SOTA methods

The proposed semi-supervised segmentation approach was benchmarked against various SOTA methods, including the mean teacher (MT (Tarvainen and Valpola, 2017)) method, consistency-training-based methods such as cross consistency training (CCT (Ouali et al.,

2020)), multi-tasking consistency (MTC (Chen et al., 2019)) and dual-task consistency (DTC (Luo et al., 2021)), pseudo-labeling based methods such as the uncertainty-aware mean teacher (UAMT(Yu et al., 2019)). We specifically selected the MT and UAMT methods due to their direct relevance to our approach. Meanwhile, MTC, CCT, and DTC were chosen to represent diverse applications of consistency training and pseudo-labeling in weakly-supervised segmentation tasks. By comparing our method against these varied approaches, we aim to provide a comprehensive perspective on its strengths. Wilcoxon signed-rank tests were employed to assess whether the performance differences were statistically significant ($p < 0.05$). All models included in the comparison were based on the same 3D U-Net with a Dropout rate of 0.15 for all convolution layers except in the output layer. The baseline method is a single 3D U-Net, trained in a fully-supervised way using only labeled images from each subset.

The model parameters were optimized using Adam optimizer (Kingma and Ba, 2014) with the initial learning rate set to $10^{-4}$ and decay rate $\beta_1$ set to 0.9 and $\beta_2$ set to 0.99. The network parameters were initialized using He initialization (He et al., 2015). The moving average decay was set to 0.99 for the teacher model. We used $k = 16$ in the nearest neighbor classification and set the number of classifiers in the ensemble to $l = 5$ (see Sec. 2.3). We ran $M = 5$ stochastic forward passes for uncertainty analysis to compute uncertainty (see Sec. 2.2). All models were trained for 40 epochs, reaching convergence as indicated by stable or increasing validation errors. And we selected the model with the lowest validation Hausdorff surface distance to evaluate the performance on the test set. The teacher input is subjected to weak Gaussian additive noise at a scale of 0.01, while the student input is exposed to more substantial Gaussian noise addictive at a scale of 0.02.

Results in Table 1 show that the proposed method outperforms the other methods in subsets in overall segmentation performance. When training on only 50 3D patches, the proposed method reaches 3.30 (mm) in HD95 and 0.929 in IoU, substantially outperforms the baseline at 9.45 in HD95 and 0.894 in IoU, and the second best semi-supervised performance (4.07 in HD95 and 0.927 in IoU from DTC). The fully-supervised method (the baseline trained on all 6266 images) reaches 0.948 IoU and 1.92 HD95.

### 3.3. Prediction Consistency and Types of Errors in the Proposed Method

This section shows the qualitative results of training the proposed semi-supervised method with 50, 150, 300, 1000, and 2500 labeled patches compared with the manual segmentation and the fully-supervised baseline. Figure 2 illustrates three challenging cases. We observe that the proposed method, only employing a limited number of labeled patches (50 and 150) in training, may produce the segmentation results (shown in the ($3^{rd}$ and $4^{th}$ columns) deviating from the reference standard ($1^{st}$ column), particularly in the presence of out of distribution samples (e.g., metal artifacts in the $2^{nd}$-row case). The proposed semi-supervised method overlooked metal artifacts (dark fan-shaped rays) caused by the in-body implants made of metal. This is because metal artifacts are uncommon in our data collection, so the chances of labeled examples containing metal artifacts are small.

The proposed method trained with limited ($\leq 300$) labeled samples may fail to capture anatomical and pathological variations, seen as over-segmentations in the $3^{rd}$ row case, covering the gap between the acetabulum and the femoral head. These regions exhibit

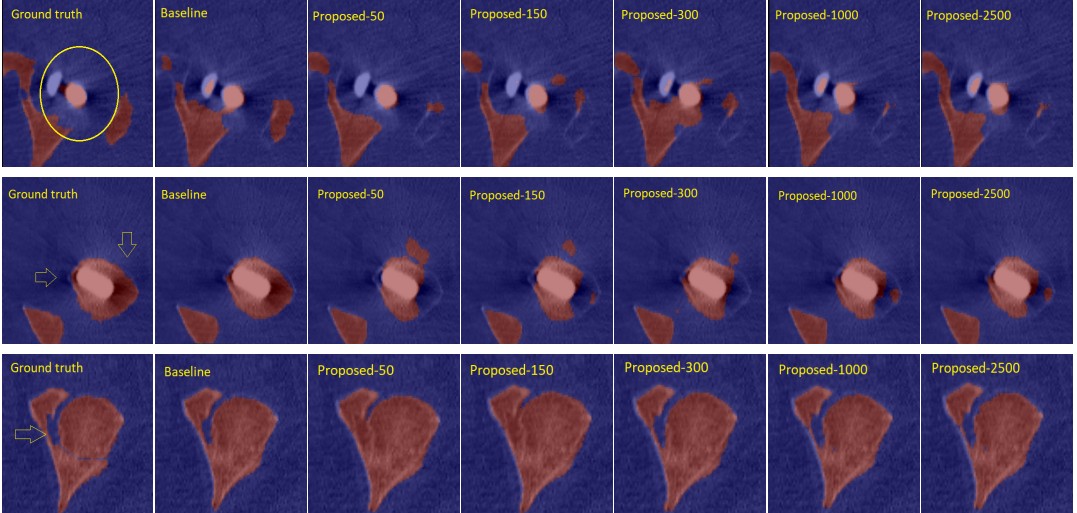

Figure 2: Qualitative results of the proposed methods trained with 50, 150, 300, 1000, and 2500 labeled patches (from the $3^{rd}$ column to $7^{th}$ column) compared with the ground truth ($1^{st}$ column) and the fully-supervised baseline ($2^{nd}$ column). We show a 3D patch using its central axial slice. By adding more labeled examples, the proposed method converges to manual labels, although still keeping consistent errors in segmenting metal artifacts ($2^{nd}$ row) and identifying local anatomical structures that are ambiguous in manual segmentation ($3^{rd}$ row).

significant anatomical and pathological variations, requiring sufficient supervision to learn powerful features. We saw that the proposed method with 1000 and more labeled images (the last two columns) performs substantially better in these regions.

## 4. Conclusion

We present a novel semi-supervised segmentation method for segmenting hip bones (pelvis and femoral head) in CT images. Our key innovation is using embedding correspondence to transfer labels from labeled voxels to unlabeled ones via the proposed neighbor matching loss and entropy minimization regularization strategy. Unlike existing methods that typically exclude voxels with unreliable predictions during pseudo-labeling, our approach can leverage these voxels to have a full coverage in the pseudo labeling process, maximizing the potential to extract valuable information from large amount of unlabeled images.

Our method outperforms other state-of-the-art techniques in hip bone segmentation from CT images, utilizing merely 50 labeled patches for training. Nonetheless, our findings highlight the necessity of a sufficient volume of labeled images to account for anatomical and pathological variations, as well as imaging artifacts. This is evidenced by the noticeable performance disparity between weakly-supervised approaches with limited labeled data and a fully-supervised method.

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

## Appendix A. Ablation Study on Loss Terms

We conducted a detailed ablation study on the key loss terms within our method: uncertainty analysis loss ($L_{UA}$), nearest neighbor matching loss ($L_{NN}$), and entropy minimization loss ($L_{EN}$). $L_{UA}$ leverages reliable teacher model predictions, while $L_{EN}$ and $L_{NN}$ address areas with less certainty, particularly around object surfaces. $L_{UA}$ was retained in all ablation scenarios. Our findings, detailed in Table 2, reveal that both $L_{EN}$ and $L_{NN}$ significantly enhance segmentation performance beyond $L_{UA}$ alone, with $L_{EN}$ showing a more pronounced individual impact by effectively optimizing embedding space separation.

Table 2: Segmentation results in IoU and HD95 of the proposed method w/o loss terms $L_{UA}$, $L_{NN}$, and $L_{EN}$ in training with 50 labeled images. Boldface denotes the result significantly better than others (p $<$0.05).

| IoU | HD95 (in mm) | $L_{UA}$ | $L_{NN}$ | $L_{EN}$ |
|------|------|------|------|------|
| 0.918 | 6.39 | ✓ | - | - |
| 0.925 | 3.94 | ✓ | ✓ | - |
| **0.929** | **3.30** | ✓ | ✓ | ✓ |
| 0.927 | 3.56 | ✓ | - | ✓ |

## Appendix B. Experiments on Hyper-Parameters for Computing Dense Similarity Maps

Several hyper-parameters are involved in the computation of dense similarity maps $K(f_{L+}^t, f_U^s)$ and $K(f_{L-}^t, f_U^s)$ as mentioned in the section 2.3.1 when measuring the unlabeled embeddings with the object embeddings and the background embeddings. This section conducts an study demonstrating their impact on the segmentation performance when trained with only 50 labeled images. The similarity measurement kernel $K$ is to compute similarity between two embedding vectors. The most common embedding similarity kernels are cosine similarity denoted as $\frac{p \cdot q}{||p||||q||}$ and euclidean similarity kernels denoted as $\frac{1}{1+||p-q||_2}$ given $q$ and $p$ are two embedding vectors. To reduce the pairwise similarity scores between $f_{L\pm}^t$ and $f_U^s$, we tested two reduce operators as either taking the maximum or the average across the $k$ labeled embeddings. As shown in Table 3, the combination of using averaging operator as the aggregation function and the cosine similarity kernel reaches the best results in terms of overall segmentation performance considering both HD95 (in mm) and IoU metrics. However, the differences among the combinations are not significant in terms of the IoU metric, while the surface distance error (HD95) is significant lower when using max operator with cosine distance. All results in this ablation study are significantly better than the SOTA methods (Table 1) in surface distance measures, showing the robustness of our method against the different choices of hyper-parameters in computing dense similarity maps.

Table 3: Segmentation results in IoU and HD95 of the proposed method using different similarity measurement kernels and reduce operators for evaluating dense similarity maps $K(f^t_{L+}, f^s_U)$ and $K(f^t_{L-}, f^s_U)$. Details refer to section 2.3.1. Boldface denotes the result significantly better than others (p $<$0.05).

| IoU | HD95 (in mm) | reduce operator | kernel |
|-----|------|-----|-----|
| 0.930 | 3.73 | mean | euclidean |
| 0.930 | 3.67 | max | euclidean |
| 0.929 | **3.30** | mean | cosine |
| 0.927 | 3.63 | max | cosine |

## Appendix C. Data collection details

950 subjects in our study underwent robotic-assisted total hip arthroplasty. All patients provided written consent allowing their data to be used for R&D purposes. Their pre-operative CT scans were analysed with proprietary software and segmented by two expert human annotators. Annotator 1 generated an initial segmentation that was reviewed, and potentially updated, by annotator 2, resulting in the final segmentation ground-truth used in this study. The annotated CT scans were utilized for both training and evaluation of our algorithm. The scans were randomly partitioned into three sets, with 350 used for training, 100 for validation, and 500 for testing. Due to the large size of CT volumes, we preprocessed all images into smaller 3D patches by randomly sampling various locations in the image with minimal overlap. The number of 3D patches for each split can be found in Table 4.

All 3D patches included in the dataset were accompanied by corresponding reference segmentations. Each of these patches underwent scoring based on the surface deviation between the manual segmentations and the segmentations derived from statistical shape models. The surface deviation was measured by the Hausdorff surface distance (with 95 % percentile, HD95) between two surfaces. Note that the segmentations produced by the statistical shape models are often influenced by common shapes. Consequently, the surface deviation serves as an indicator of the anatomical variation present in relation to these common shapes, thereby reflecting the complexity of the segmentation. The score associated with surface deviation is referred to as the surface deviation score (SDS). SDS scores can further be categorized into six groups (ranging from 0 to 5). Table 4 provides an overview of the distribution of surface deviation scores in each group across the training, validation, and test sets. Specifically, the training set consists of 6266 3D patches, while the validation and test sets contain 1174 and 1200 3D patches, respectively. There is a relatively smaller number of 3D patches in the test set (n=1200). This is because we sampled 200 3D patches from each of the six pre-defined SDS groups. The mean and standard deviation of SDS scores from each group can be found in Table 4, measured in HD95. We collected SDS scores to show that our dataset consists of many challenging cases as they have high degree of shape deformations, i.e., possibly either a heavily-diseased case or representing other complexities (e.g. metal artifacts) introduced by already-placed implants.

Table 4: Train, validation, and test split in our data collection $i$th 3D patches sampled from 950 CT scans. We show the distribution of surface deviation scores (SDS) across the splits. SDS scores are categorized into six-ordinal scales (ranging from 0 to 5) and we show the statistics (measured in HD95 with mean±standard deviation) of SDS scores for each category.

| SDS | HD95 (mm) | # train 3D patches | # valid 3D patches | # test 3D patches |
|---|---|---|---|---|
| 0 | 0.99±0.03 | 855 | 47 | 200 |
| 1 | 1.41±1.28 | 1641 | 239 | 200 |
| 2 | 2.04±0.21 | 2291 | 455 | 200 |
| 3 | 3.42±0.53 | 972 | 290 | 200 |
| 4 | 5.90±0.84 | 325 | 104 | 200 |
| 5 | 10.57±1.93 | 182 | 39 | 200 |
| total | | 6266 | 1174 | 1200 |

