# OpenReview forum: "Semi-Supervised Segmentation via Embedding Matching"
_MIDL.io/2024/Conference — MIDL 2024 Oral_

### Official Review · Reviewer_P1cS · 2024-02-26

**Confidence:** 5
**Preliminary Rating:** 4
**Recommendation:** Oral
**Final Rating:** 5

**Summary:**

The paper presents a semi-supervised segmentation scheme utilizing a teacher-student architecture combined with an embedding matching methodology. The proposed framework can sufficiently employ unlabeled images to enhance segmentation performance. In this methodology, the teacher’s segmentation results with high uncertainty are improved by a voxel-wise embedding correspondence. Evaluation experiments and comparisons indicate that the proposed method slightly outperformed the selected state-of-the-art methods.

**Strengths:**

The paper is well structured with appropriate usage of English grammar and syntax. The authors presented related works on semi-supervised segmentation methods in general and in more specific methods with perturbations and pseudo-labeling.
The methodology is adequately presented with sufficient details. Thorough comparison experiments with state-of-the-art semi-supervised techniques have been conducted.

**Weaknesses:**

The proposed method’s results are quite close to the compared methods even for the 50 patches training (Table 1). This indicates that the model’s accuracy is approximately similar to the other state-of-the-art methods that were compared.

**Detailed Comments:**

Due to the similarity of the results and the small superiority of the proposed method in the comparisons, presenting its advantages against the compared methods would enhance the presentation of the paper’s contribution.

**Justification Of Final Rating:**

The article presents a novel semi-supervised hip bone segmentation methodology utilizing a teacher-student architecture combined with an embedding matching strategy. The methodology and theoretical background are well described. The authors presented adequately the main contributions and comparisons against the state-of-the-art methods. Extensive evaluation experiments and ablation studies support the superiority of the method. Therefore, my recommendation is Strong Accept.

**Justification Of The Preliminary Rating:**

The methodology is sufficiently presented and described throughout the paper. Details on the evaluation experiments such as the choice of the number of epochs should be added. The results of the proposed method in terms of DSC and HD are quite close to the compared methods which could be further analyzed. Clearly stating contributions against the compared methods could support the proposed method.

**Questions To Address In The Rebuttal:**

Please provide additional details on the choice of the number of epochs during experiments as stated in the “Questions” section. The authors should justify the choice of this number and if this number of epochs is adequate for the training of all models until convergence. Due to the variant methodologies to be compared, the same number of epochs may not be sufficient to train all the models leading to underperformance.

---

> ### Author Response · Authors · 2024-03-17
> **Two ablation studies added, statistical tests were conducted.**
>
> We are grateful for the reviewer's comments on the strengths and weaknesses of our paper. In our revision, we have added two ablation studies to show how our algorithm performs with various hyper-parameters, as well as the combination of loss terms. Below, we provide detailed responses to each of the concerns raised. We start with the origin reviewer's comment, followed by our response.
> - The proposed method’s results are quite close to the compared methods even for the 50 patches training (Table 1). This indicates that the model’s accuracy is approximately similar to the other state-of-the-art methods that were compared.
>
> We have conducted statistical tests in our revision (mentioned in section 3.2) to demonstrate the significance of our improvement, especially when leveraging only 50 labeled images in training.
> - Please provide additional details on the choice of the number of epochs during experiments as stated in the “Questions” section. The authors should justify the choice of this number and if this number of epochs is adequate for the training of all models until convergence. Due to the variant methodologies to be compared, the same number of epochs may not be sufficient to train all the models leading to underperformance.
>
> Thank you for your request for further clarification regarding our choice of the number of epochs in our experiments. In our study, we carefully monitored the convergence of all models throughout the training process. We determined that the number of epochs selected was sufficient to ensure that each model was well-trained and reached convergence. We stopped our training when the validation errors do not show changes or started increasing.  We added the discussion in section 3.2 in the revision.

---

### Official Review · Reviewer_xH3R · 2024-03-02

**Confidence:** 4
**Preliminary Rating:** 2
**Final Rating:** 4

**Summary:**

The paper introduces a semi-supervised segmentation method for hip bones in CT images, utilizing embedding correspondence and neighbor matching loss to propagate labels from labeled to unlabeled voxels. The authors claim superior performance over existing state-of-the-art methods, especially in scenarios with limited labeled data.

**Strengths:**

The paper is well-written and easily comprehensible. It contributes to the field by providing a thorough comparison with several state-of-the-art methods, elucidating the efficacy of the proposed approach in scenarios with sparse labeled data.

**Weaknesses:**

Supervised performance on 2500 patches across 180 CT scans is very similar to semi-supervised performance on the entire dataset (350 CT scans). Even for 71 CT scans the performance is possibly equivalent. So, it is unclear how much this approach is practically relevant.

The current evaluation lacks confidence intervals or statistical tests which makes it harder to understand the significance of the performance improvements.

It would be great to get more clarity on the challenges of segmenting hip bones to better estimate the significance of this work.

The paper appears to incorporate disparate approaches without establishing a cohesive motivation and narrative. The authors distinguish embeddings from object and background pixels, create an ensemble of nearest neighbor classifiers, add an entropy minimization loss term, and apply different noise for student and teacher. Why is each of these steps needed and what is the merit of the individual steps?

**Detailed Comments:**

No further comments.

**Justification Of Final Rating:**

The authors addressed several concerns in their rebuttal. Based on these responses, I have changed the final rating. As written in the response, I still believe that the manuscript has limited practical relevance and could be evaluated on public datasets.

**Justification Of The Preliminary Rating:**

The authors propose a semi-supervised segmentation method for hip bones in CT images, but the practical relevance of the approach is questionable due to similar performance between supervised and semi-supervised methods across different dataset sizes. Additionally, the absence of confidence intervals, the ease of segmenting hip bones, and the lack of explanation for the effectiveness of the embedding approach all contribute to the rating. The presentation seems fragmented and ablation studies are missing.

**Questions To Address In The Rebuttal:**

1) Could you provide ablation studies that motivate the necessity of each of the individual steps?

2) What is the practical relevance of this approach given that on 180 CT scans, there seems to be no advantage compared to the semi-supervised approach? Segmenting 180 CT scans seems to be feasible, e.g., using prompt-based image segmentation methods like MedSAM.

3) Could you check for the statistical significance of your results with statistical tests?

**Special Issue:**

No

---

> ### Author Response · Authors · 2024-03-17
> **Two ablation studies added, conducted statistical tests.**
>
> We are grateful for the reviewer's constructive feedback. Below, we provide detailed responses to each of the concerns raised. We start with the origin reviewer's comment, followed by our response.
>
> - Supervised performance on 2500 patches across 180 CT scans is very similar to semi-supervised performance on the entire dataset (350 CT scans). Even for 71 CT scans the performance is possibly equivalent. So, it is unclear how much this approach is practically relevant.
>
> We thank the reviewer for their insightful comments on our performance comparison between semi-supervised and fully supervised approaches. Our study focuses on demonstrating the effectiveness of our method in enhancing segmentation with far fewer labeled examples, a critical asset given the dataset's complexity and the substantial labeling effort required. To underscore the dataset's challenging nature, we've included a table (Table 4) in the appendix detailing case complexity scores. This highlights the task's inherent difficulties, particularly in the context of total hip arthroplasty, where pathological changes and imaging artifacts make extensive labeling both costly and time-consuming. Thus, our method's ability to achieve improved segmentation with limited labeled data is not only significant but also highly relevant for practical applications.
>
> - The current evaluation lacks confidence intervals or statistical tests which makes it harder to understand the significance of the performance improvements.
>
> In the revision, a Wilcoxon signed-rank test was employed to assess whether the performance difference was statistically significant (p $<$ 0.05). We mentioned this in section 2.3.
>
> - It would be great to get more clarity on the challenges of segmenting hip bones to better estimate the significance of this work.
>
> To highlight the complexity of our dataset, we've added Table 4 to the appendix, presenting case complexity score distributions. This illuminates the inherent challenges in segmenting hip joints for total hip arthroplasty, where pathological variations and imaging artifacts, especially from pre-existing metal implants, significantly complicate the task.
>
> - The paper appears to incorporate disparate approaches without establishing a cohesive motivation and narrative. The authors distinguish embeddings from object and background pixels, create an ensemble of nearest neighbor classifiers, add an entropy minimization loss term, and apply different noise for student and teacher. Why is each of these steps needed and what is the merit of the individual steps?
>
> In the revised manuscript, we've incorporated two ablation studies in the appendix (Table 2 and 3) to assess the impact of various loss terms and the hyper-parameters involved in computing dense similarity maps.
>
> Questions To Address In The Rebuttal:
> - Could you provide ablation studies that motivate the necessity of each of the individual steps?
>
> Added to the appendix.
> - What is the practical relevance of this approach given that on 180 CT scans, there seems to be no advantage compared to the semi-supervised approach? Segmenting 180 CT scans seems to be feasible, e.g., using prompt-based image segmentation methods like MedSAM.
>
> We sincerely appreciate your insightful inquiry regarding the practical relevance of our approach, particularly in the context of the results obtained from 180 CT scans. Our method aims to demonstrate the significant advantages of semi-supervised learning in scenarios where labeled data is scarce or costly to obtain.
> We acknowledge that prompt-based segmentation methods like MedSAM could be effective for some applications. However, these heavy methods are often computationally intensive, requiring at least 6GB of GPU memory for processing a single 2D image at 1024x1024 resolution, and the inference process is slow (> 6 seconds per 2D slice) on commonly available hardware based on our own experience. Moreover, applying SAM or MedSAM introduces new challenges, such as boundary ambiguity, which may necessitate multiple points or boxes prompts from the user to accurately define object contours, further increasing the time and effort required for prompting.
>
> - Could you check for the statistical significance of your results with statistical tests?
>
> We've incorporated statistical test results for all the tables presented in the revised manuscript. Thank you for this valuable suggestion.

---

> > ### Comment · Reviewer_xH3R · 2024-03-27
> >
> > I appreciate the detailed answers to my questions and the inclusion of the statistical test. Based on these updates, I have revised my rating to "weak accept." Although I understand the limitations of seeing improvements
> > for only a small number of images, the appendix reveals a substantial availability of subjects, 950 to be exact, which would enable the training of a fully supervised algorithm that is comparable to the proposed approach.
> > Consequently, in my view, the practical relevance remains limited.
> >
> > The potential withholding of the source code creates significant obstacles for the scientific community to reproduce and delve into the intricacies of the algorithm. To address this issue, an evaluation on public datasets
> > would be highly beneficial. It is possible that the private dataset is more complete or comprehensible, but I still believe that evaluation on a public dataset is indispensable, particularly if the code may not be made
> > available.

---

> ### Author Response · Authors · 2024-03-28
> **source code available**
>
> we are agreed to share the source code upon acceptance.

---

### Official Review · Reviewer_nv7C · 2024-03-07

**Confidence:** 4
**Preliminary Rating:** 4
**Recommendation:** Poster
**Final Rating:** 4

**Summary:**

This paper presents an innovative semi-supervised method for semantic segmentation of hip bones in CT images. The novelty lies in the pseudo labeling, with propagation from labeled to unlabeled images through voxel-wise matching of embeddings, providing a fresh perspective on utilizing pseudo labels even if the prediction from the teacher is unreliable. The authors assess the performance of their model on a private cohort of 950 patients with CT scans, outperforming state-of-the-art semi-supervised methods while not surpassing a fully supervised approach.

**Strengths:**

Basic considerations:
- The paper is well written, well structured, and is easy to follow.
- Solid mathematical modeling of solving the challenge, with proper loss equations.

Other:
- The methodological contribution for label propagation makes sense both from an ML-related (get closer to supervised setting with more annotated data, and work at the last-layer embedding level) and clinical (focus on surface and boundaries where annotations can be blurry) perspective.
- Good knowledge of the literature, particularly the Mean Teacher approach for consistency training and uncertainty analysis for pseudo-labeling. The authors propose a new iteration in a semi-supervised setting, leveraging and implementing popular methods and inserting a new (missing) block to improve global performance.
- The results are promising with improvements compared to previous methods, and an interesting qualitative analysis is performed on successful and failed cases.

**Weaknesses:**

- Even if the novelty is undeniable, one could question its magnitude since only subsection 2.3 is innovative. 2.2 is a simple repeat of [1], and the moving average is the same as [2].
- The performance is only assessed on one (private) dataset, with a very specific location and task. It would be interesting to see how the model behaves in other locations, and it is easy to test it with public datasets since the implementation is rather versatile.
- Source code sharing would be a plus for reproducibility.
- Basic statistical analysis with standard deviation for IoU and HD95 is required to truly assess the significance of the results.
- Ablative studies would have been really interesting: the entropy minimization loss makes sense, and its role is well explained, but the authors could have assessed its importance (or relevance?) by playing with a different weight from the NN matching loss. Anyway, some comments would have been appreciated about the relative importance of each component, the choice of ramp-up hyperparameters, the number of sampled voxels k, or the number of classifiers l (I am not referring to usual hyperparameters like initialization, weight decay or dropout rate). Such ablative studies could have been presented with curves or additional rows in the results table.
-	The choice of SOTA methods needs more justification since the author cites other papers in their review of the literature
-	The main Figure 1 is not clear enough, I would suggest using different categories of boxes depending on the object (model, feature, loss, measure, image, …).

**Detailed Comments:**

- Clarifications are needed about the manual segmentation with proprietary software. Is there only one annotator? How many years of experience?
- Please be precise about the sampling procedure, especially how you ended up with 5 times fewer patches in the testing set than in the training set while you had more patients.
- Please explain with quantitative arguments the sentence “Interestingly, even when the labeled embeddings are on the wrong side of the true decision hyperplane, and the teacher network mislabels their corresponding voxels, we can still utilize the label of these voxels to guide the student training of other voxels in the unlabeled images given their embeddings similarity”

[1] Lequan Yu, Shujun Wang, Xiaomeng Li, Chi-Wing Fu, and Pheng-Ann Heng. Uncertaintyaware self-ensembling model for semi-supervised 3d left atrium segmentation. In Medical Image Computing and Computer Assisted Intervention–MICCAI 2019: 22nd International Conference, Shenzhen, China, October 13–17, 2019, Proceedings, Part II 22, pages 605–613. Springer, 2019.

[2] Antti Tarvainen and Harri Valpola. Mean teachers are better role models: Weight-averaged consistency targets improve semi-supervised deep learning results. Advances in neural information processing systems, 30, 2017.

**Justification Of Final Rating:**

Authors have addressed most of the concerns raised in the first review with ablative studies and statistical tests. The methodology is interesting and contributes to the field, even if additional experiments/datasets would add value to the paper. I recommend weak accept.

**Justification Of The Preliminary Rating:**

The paper is interesting with a methodological novelty for a semi-supervised setting and the use of images for which the teacher cannot provide reliable predictions. The mathematical formulation is clear and solid. More details about the results and design choices, as well as better generalization and reproducibility would increase the quality of the study. Therefore, I recommend weak acceptance.

**Questions To Address In The Rebuttal:**

- The authors mention that they provide supplemental information to assess the choice of the similarity measurement and reduce operations, which is a very sound idea, but no such material is available.
- More detailed quantitative analysis of results should be tackled in the rebuttal, from ablative studies to benchmark choices and statistical significance.
- Discussion about the robustness of the approach for other locations and tasks would be highly appreciated.

**Special Issue:**

No

---

> ### Author Response · Authors · 2024-03-17
> **Ablation studies and statistical tests added**
>
> We are grateful for the reviewer's acknowledgment of our paper's novelty and appreciate the constructive feedback. Below, we provide detailed responses to each of the concerns raised. We start with the origin reviewer's comment, followed by our response.
>
> - Even if the novelty is undeniable, one could question its magnitude since only subsection 2.3 is innovative. 2.2 is a simple repeat of [1], and the moving average is the same as [2].
>
> Indeed, the core innovation of our work is concentrated within section 2.3, where we introduce a novel approach that leverages embedding analysis for the pseudo-labeling of regions or voxels. We view this as a foundational contribution, offering a versatile framework that enriches the current landscape of semi-supervised learning methodologies.
> - The performance is only assessed on one (private) dataset, with a very specific location and task. It would be interesting to see how the model behaves in other locations, and it is easy to test it with public datasets since the implementation is rather versatile.
>
> We understand and appreciate the reviewer's suggestion to validate our approach across multiple datasets to enhance the generalizability of our findings. However, our decision to focus on a single private dataset was driven by following considerations.
> Our dataset with curated manual segmentations, was carefully compiled to include a diverse array of pathologies, anatomical variations, and imaging artifacts in hip bone CT images, crucial for thoroughly testing the effectiveness and robustness of our segmentation approach. In the appendix (Table 4), we present the distribution of case complexity scores within our dataset to highlight the segmentation challenges, illustrating the diverse conditions our method is designed to address.
>
> - Source code sharing would be a plus for reproducibility.
>
> We are actively considering the possibility of sharing our source code upon acceptance and are currently in discussions regarding this matter. The final decision will adhere to our scientific dissemination review protocol, in line with the relevant company policies.
>
> - Basic statistical analysis with standard deviation for IoU and HD95 is required to truly assess the significance of the results.
>
> We employed a Wilcoxon signed-rank test (p < 0.05) to determine if the performance differences observed were statistically significant. We mentioned this in section 3.2.
>
> - Ablative studies would have been really interesting: the entropy minimization loss makes sense, and its role is well explained, but the authors could have assessed its importance (or relevance?) by playing with a different weight from the NN matching loss.
>
> In the appendix, we've incorporated two ablation studies. The first assesses the significance of each loss term, including entropy loss, nearest neighbor matching loss, and uncertainty analysis loss. The second examines the effects of varying similarity measurement functions and reduction operations on dense similarity map computation. We consider these elements crucial to our model's design. Our ramp-up parameters follow the precisely defined ramp-up function from the uncertainty-aware mean teacher approach. The choice of the number of sampled voxels, k, is constrained by computational resources, with a larger k offering diminishing returns due to the computational demand and the randomness in sampling voxels.
>
> - The choice of SOTA methods needs more justification since the author cites other papers in their review of the literature.
>
> MT and UAMT were selected because they are the most relevant to our method. MTC, CCT and DTC represents different way of using consistency training and pseudo labeling in weakly-supervised segmentation. We believe bringing methods in different lines of works in comparison provides comprehensive views of the strength of the proposed method. We added discussions in section 3.2.
> - The main Figure 1 is not clear enough, I would suggest using different categories of boxes depending on the object (model, feature, loss, measure, image, …).
>
> We have updated Figure 1 to differentiate models and features through distinct color coding.
>
> Questions:
> - The authors mention that they provide supplemental information to assess the choice of the similarity measurement and reduce operations, which is a very sound idea, but no such material is available.
> Added.
> - More detailed quantitative analysis of results should be tackled in the rebuttal, from ablative studies to benchmark choices and statistical significance.
> Added.
> - Discussion about the robustness of the approach for other locations and tasks would be highly appreciated.
> Up to this point, our efforts have been concentrated on conducting a comprehensive analysis of the dataset presented in our study. We believe our future work will allow us to provide the empirical evidence necessary to fully demonstrate the versatility of our approach across various segmentation problems.

---

### Author Response · Authors · 2024-03-17
**Ablation studies are added, minor changes are made as suggested by reviewers, details in our data collection are provided.**

We sincerely thank the reviewers for their insightful and constructive feedback, which has undoubtedly helped improve the quality and clarity of our manuscript. We have carefully considered each comment and have made substantial revisions to address these concerns. Below, we summarize our responses to the main points raised, demonstrating how we have incorporated the feedback into our revised manuscript.

Ablation studies are added in the appendix of the revised manuscript.
 - The first ablation study (Table 2) shows that the proposed method w/o entropy loss, w/o nearest neighbor matching loss in combination with the uncertainty analysis loss.
 - The second ablation study (Table 3) meticulously examines the performance of our method across different embedding similarity functions and reduction operators used to compute distance similarity maps. These maps are crucial for determining nearest neighbor matching loss and entropy loss, underscoring their importance in our methodology.

We have also conducted statistical analyses, specifically the Wilcoxon signed-rank test (p < 0.05) mentioned in section 2.3, across all reported results to rigorously determine the significance of the improvements observed.

We have provided a comprehensive explanation of our data collection protocols in the appendix (Table 4), detailing the acquisition of reference segmentations and the generation process for the 3D patches utilized in the training and validation phases of our method.

We have implemented minor revisions to enhance the manuscript's quality further. These adjustments, along with the clarifications provided, aim to fully address the reviewers' concerns and significantly elevate our contribution to the domains of semi-supervised learning and medical image segmentation. We are thankful for the chance to refine our work in light of the insightful feedback received and are hopeful that our revised manuscript aligns with the conference's acceptance criteria.

---

### Comment · Area_Chair_1K86 · 2024-03-18
**Please read and respond to author comments**

Dear reviewers. The authors have posted responses to your reviews. Please take the time to read and respond before March 27.

---

### Meta-Review · Area_Chair_1K86 · 2024-04-02

**Recommendation:** Accept (Oral)
**Confidence:** 5

**Metareview:**

The authors propose a novel semi-supervised segmentation method and apply it to hip bone segmentation. The reviewers found the idea of the paper innovative with strong mathematical modeling and promising results. Inclusion of ablation studies and statistical results in the rebuttal resulted in reviewers raising their ratings resulting in one strong accept and two weak accepts.

---

### Decision · Program_Chairs · 2024-04-05

Accept (Oral)

---

> ### Author Response · Authors · 2024-04-08
> **Oral or poster presentation**
>
> Dear Program Chairs,
>
> Thank you for your diligence and effort in the review process of our manuscript. Could you please inform us whether the paper has been accepted for an oral or a poster presentation?
>
> Best, weiyi